# Distribution of ascariasis, trichuriasis and hookworm infections in Ogun State, Southwestern Nigeria

Hammed Oladeji Mogaji[1]*, Gabriel Adewunmi Dedeke[2], Babatunde Saheed Bada[3], Samuel Bankole[2], Adejuwon Adeniji[2], Mariam Tobi Fagbenro[2], Olaitan Olamide Omitola[2], Akinola Stephen Oluwole[4], Nnayere Simon Odoemene[5], Eniola Micheal Abe[6], Chiedu Felix Mafiana[7], Uwem Friday Ekpo[2]

1 Department of Animal and Environmental Biology, Federal University Oye-Ekiti, Ekiti, Nigeria,
2 Department of Pure and Applied Zoology, Federal University of Agriculture, Abeokuta, Nigeria,
3 Department of Environmental Management and Toxicology, Federal University of Agriculture, Abeokuta, Nigeria, 4 COUNTDOWN Project, Sightsavers, Nigeria Country Office, Kaduna, Nigeria, 5 Department of Biological Sciences, Adeleke University, Ede, Osun State, Nigeria, 6 National Institute of Parasitic Disease and Control, 14 China Centre for Disease Control, Beijing, P.R. China, 7 Directorate for Research and Innovation, National Open University of Nigeria, Abuja, Nigeria

* mogajihammed@gmail.com

**Data Availability Statement:** All relevant data are within the paper and its Supporting Information files.

## Abstract

### Background

Ascariasis, Trichuriasis and Hookworm infections poses a considerable public health burden in Sub-Saharan Africa, and a sound understanding of their spatial distribution facilitates to better target control interventions. This study, therefore, assessed the prevalence of the trio, and mapped their spatial distribution in the 20 administrative regions of Ogun State, Nigeria.

### Methods

Parasitological surveys were carried out in 1,499 households across 33 spatially selected communities. Fresh stool samples were collected from 1,027 consenting participants and processed using ether concentration method. The locations of the communities were georeferenced using a GPS device while demographic data were obtained using a standardized form. Data were analysed using SPSS software and visualizations and plotting maps were made in ArcGIS software.

### Results

Findings showed that 19 of the 20 regions were endemic for one or more kind of the three infections, with an aggregated prevalence of 17.2%. Ascariasis was the most frequently observed parasitic infection in 28 communities with a prevalence of 13.6%, followed by hookworm infections with a prevalence of 4.6% while Trichuriasis was the least encountered with a prevalence of 1.7%. The spatial distribution of infections ranges between 5.3–49.2% across the regions. The highest and lowest distribution of overall helminth infections was

**Funding:** The author(s) received no specific funding for this work.

**Competing interests:** The authors have declared that no competing interests exist.

**Abbreviations:** GE, Google Earth; GPS, Geographic Positioning System; LGAs, Local Government Areas; SAF, Sodium Acetate Acetic Acid Formaladehyde; SPSS, Statistical Package for Social Sciences; STH, Soil Transmitted Helminths; WHO, World Health Organization.

recorded in Egbado South and Egbado North respectively. Nine regions had infection status between 20.0%-49.2%, while 10 regions had infection status between 5.3%-15.8%.

## Conclusion

This study provides epidemiological data on the prevalence and spatial distribution of ascariasis, trichuriasis and hookworm infections which will add to the baseline data and guide the public health officers in providing appropriate control strategies in the endemic communities.

## Background

Infections with *Ascaris lumbricoides* (Ascariasis), *Trichuris trichiura* (Trichuriasis) and hookworms (Ancyclostomiasis / Necatoriasis) are widely distributed in tropical and subtropical areas, with the greatest numbers occurring in sub-Saharan Africa, the Americas, China and East Asia [1]. Transmission of these parasites is prominent in areas with favorable climatic and environmental conditions, poor access to potable water supply, sanitation and hygiene resources [2]. Recent estimates shows that more than 5 billion people are at risk, and about 1.5 billion people are currently infected [1]. Besides, about 267 million pre-schoolers (age 2–5 years) and 568 million school-aged children (age 5–14 years) who live in areas where these parasites are intensively transmitted are at risk [3]. These children suffer the major brunt of these infections, with leading clinical manifestations such as malnutrition and iron deficiency anemia [4]. In addition, children co-infected with these parasites show impaired cognitive and physical development leading to significant reductions in educational gains via inefficient learning and other school achievements [5,6]. Nonetheless, in most endemic settings, there is an established morbidity control programme targeted at school-aged children due to high worm burden as a result of their frequent contact with contaminated soils, and exposure to poor sanitation and unhygienic conditions [7,8]. The World Health Organization through this programme recommends large-scale administration of albendazole, either once a year (annually) when the baseline prevalence of infections is between 20 and 50%, or twice a year (biannually) when the prevalence is above 50% [7]. This strategy costs donor agencies and developing economies billions of dollar every year. However, in resource challenged settings where there are records of sporadic large-scale administration of anthelmintic drugs, there is need to constantly delineate endemic areas, and provide robust maps to identify hot spots. Such maps are essential to facilitate better targeting and efficient delivery of cost effective control interventions [9]. This study therefore mapped the spatial distribution of Ascariasis, Trichuriasis and Hookworm infections in Ogun State, southwestern Nigeria.

## Methods

### Study area

Ogun State is situated in the southwestern part of Nigeria and is made up of 20 administrative local government areas (LGAs) with Abeokuta as the capital city. The state has a landmass of 16,085 km$^2$, and is located within longitude 2°45$^1$E and 3°55$^1$E and latitude 7°01$^1$N and 7°18$^1$N. The State is highly urbanized, with a population estimate of about 5million inhabitants, an annual growth rate of 2.83%, and about 95% of the inhabitants are of the Yoruba tribe. The State covers a wide range of vegetation zones, from the freshwater swamp with mangrove

forest in the southeast, through diverse forest communities to the woody guinea savannah in the northwestern tip of the state. The rain forest is the largest ecological zone running through the centre of the state from east to west. Annual rainfall ranges from 900mm in the northern parts, up to 1600 mm along the coast. Major occupations of the population are farming, timber logging, and trading. Primary schools exist in most communities but in some cases, two or more communities share the same school.

## Study design and sampling procedures

This is a cross-sectional and community-based study, conducted between July 2016 and November 2018. Systematic point sampling method was employed in selection of sampling sites to ensure an unbiased and fair representation of communities across the 20 administrative LGAs in the State. As an initial step in the selection process, a 15 km x 15 km sized grid was placed on the administrative map of Ogun state in Google Earth (GE) software using the GE path tool. The centre of each grid was located in Arc GIS 9.3 software, and the geographical coordinates were recorded. The closest community to the centre of each grid was identified and selected using Google Earth software. A total of 33 communities were selected across the 20 LGAs in the state as the study communities (S1 Dataset).

## Selection of households for survey

A total sampling method was employed for household selection in the communities surveyed. Prior to data collection, selected communities were visited and with the permission of the community leaders, meetings were held with the members of the community to intimate them about the purpose of the study and the procedures to be adopted. Every member of the household is eligible for participating in the research. In each community, a house, usually at the centre of the community was designated as an area of work for processing and microscopic examination of stool samples. The location of the communities was georeferenced with Garmin 20.0 GPS device.

## Ethical approval and consent to participate

Ethical clearance for this study (HPRS/381/183) was obtained from the Ethics review committee of Department of Planning, Research and Statistics, Ogun State Ministry of Health, Oke Imosan Abeokuta, Nigeria. For each household visited, consent forms were made available to household members after explaining the objectives of the research to them. Members willing to participate in the research and who completed written consent froms were enrolled into the study. However, for children below age sixteen, consent was provided by their parents after completing an assent form on their behalf.

## Faecal samples collections and processing

A stool container was distributed to consenting participants in each of the georeferenced household. Participants' unique identifiers were marked on the containers and detailed instructions of how to collect a fresh morning stool sample were given. Individuals who consented to take part in the study were requested to provide stool up to half (about 5gm) into the labelled plastic bottles provided. Stool samples were processed using the SAF- Ether (Sodium acetate-acetic acid formalin—ether) concentration method in the designated area of work within the community. One gram (1g) of the faecal sample was emulsified in already prepared 10 ml of SAF in another sample bottle. The bottle was covered and shaken vigorously to form a cloudy suspension. The stool suspension was strained into a centrifuge tube using double

gauze of about 13 mm diameter placed in a funnel. The residue was discarded while the filtrate was centrifuged at 2000 rpm for 1 minute. The supernatant was decanted. 7 ml of normal saline was later added to the sediment, after which 3 ml of ether was finally added to the suspension [10]. A stopper was placed on the tube, and the mixture was shaken vigorously before centrifuging for 5 minutes at 2000 rpm. The first three layers of the suspension observed after centrifuging was discarded, leaving the last layer of sediment. This sediment was then pipetted on a clean microscopic slide. Two slides were prepared from 1g of each stool sample by experienced laboratory technicians 2 hours post sample collection [10].

### Estimation of parasite's prevalence and intensity

Prepared slides were examined under a compound microscope for microscopic ova or larva of the three parasites (*Ascaris spp.*, *Trichuris spp.* and Hookworms). The parasites' eggs were counted for each species, and number of eggs per species and per stool examined was recorded for each participant. Mean infection intensity estimates were computed for each examined person on a logarithmic scale for the purpose of data normalization i.e. EPG = Log (n +1), where EPG means egg per gram.

### Data management, analysis and visualizations

Data obtained during the survey i.e. demographics, were first subjected to descriptive analysis in SPSS 20.0 software, and results were reported in frequencies and percentages. Prevalence and intensity estimates of helminth infections were also computed for each of the communities surveyed and reported accordingly. Significance level was set at P≤0.05. Data were imported into ArcGIS 9.3 software for visualizations and plotting maps.

## Results

### Sex and age distribution of study participants

A total of 1499 participants comprising of 899 (60%) females and 600 (40%) males were enrolled across the 33 communities in 20LGAs out of which 1024 (68.5%) returned stool samples for laboratory analysis. The sex ratio of females to males was 3:2. The ages of the enrolled participants are provided in Table 1.

### Spatial distribution of *Ascaris, Trichuris* and hookworm infections in Ogun State

Fig 1 shows the spatial distribution of the three parasites' in Ogun State. Of the 20 LGAs examined, 19(95.0%) were endemic for one or more of the three parasite. *Ascaris lumbricoides* are the most geographically distributed, found in 28(84.8%) out of the 33 communities and 19 (95.0%) out of 20LGAs. Hookworm was present in 19(57.6%) out of 33 communities and 15 (75.0%) out of 20LGAs. *Trichuris trichiura* infections were found in 9(27.3%) out of 33 communities and 7(35.0%) out of 20LGAs.

### Co-distribution patterns of *Ascaris, Trichuris* and hookworm infections in Ogun State

Fig 2 shows the co-distribution patterns of STH infections in the State. Single infection was recorded for *Ascaris lumbricoides* in 3 LGAs; Ewekoro, Ikenne and Ijebu northeast. For double infections, combination of *Ascaris* and Hookworm were the most predominant, observed in 15 LGAs except Ewekoro, Ijebu east, Ijebu northeast, Ikenne and Ijebu ode. Combination of

**Table 1. Sex and age distribution of study participants.**

| SN | LGA | Number Examined | Sex (%) | | Age in years (%) | | | | | |
|----|-----|-----------------|---------|---------|---------|---------|---------|---------|---------|---------|
| | | | Male | Female | <5yrs | 5-15yrs | 16-25yrs | 26-40yrs | 41-70yrs | >70yrs |
| 1 | Abeokuta north | 87 | 28(32.2) | 59(67.8) | 7(8.0) | 64(7.6) | 9(10.3) | 3(3.4) | 4(4.6) | 0(0) |
| 2 | Abeokuta south | 20 | 11(55.0) | 9(45.0) | 0(0) | 0(0) | 6(30.0) | 4(20.0) | 10(50.0) | 0(0) |
| 3 | Ado-odo ota | 84 | 32(38.1) | 52(61.9) | 0(0) | 1(1.2) | 12(14.3) | 23(27.4) | 43(51.2) | 5(6.0) |
| 4 | Ewekoro | 61 | 23(37.7) | 38(62.3) | 0(0) | 21(34.4) | 10(16.4) | 15(24.6) | 15(24.6) | 0(0) |
| 5 | Ifo | 37 | 13(35.1) | 24(64.9) | 0(0) | 0(0) | 3(8.1) | 17(45.9) | 15(40.5) | 2(5.4) |
| 6 | Ijebu east | 71 | 34(47.9) | 37(52.1) | 0(0) | 2(2.8) | 12(16.9) | 23(32.4) | 30(42.3) | 4(5.6) |
| 7 | Ijebu north | 54 | 35(64.8) | 19(35.2) | 0(0) | 1(1.9) | 2(3.7) | 17(31.5) | 23(42.6) | 11(20.4) |
| 8 | Ijebu north-east | 27 | 15(55.6) | 12(44.4) | 0(0) | 0(0) | 0(0) | 4(14.8) | 12(44.4) | 11(40.7) |
| 9 | Ijebu ode | 46 | 17(37.0) | 29(63.0) | 0(0) | 0(0) | 6(13.0) | 23(50.0) | 1(28.3) | 4(8.7) |
| 10 | Ikenne | 39 | 18(46.2) | 21(53.8) | 1(2.6) | 13(33.3) | 4(10.3) | 7(17.9) | 10(25.6) | 4(10.3) |
| 11 | Imeko-afon | 90 | 31(34.4) | 59(65.6) | 1(1.1) | 17(18.9) | 16(17.8) | 28(31.1) | 23(25.6) | 5(5.6) |
| 12 | Ipokia | 42 | 20(47.6) | 22(52.4) | 0(0) | 1(2.4) | 9(21.4) | 17(40.5) | 14(33.3) | 1(2.4) |
| 13 | Obafemi-owode | 89 | 35(39.3) | 54(60.7) | 0(0) | 1(1.1) | 16(18.0) | 34(38.2) | 31(34.8) | 7(7.9) |
| 14 | Odeda | 108 | 54(50.0) | 54(50.0) | 0(0) | 4(3.7) | 10(9.3) | 37(34.3) | 45(41.7) | 12(11.1) |
| 15 | Odogbolu | 77 | 22(28.6) | 55(71.4) | 0(0) | 3(3.9) | 3(3.9) | 21(27.3) | 41(53.2) | 9(11.7) |
| 16 | Ogun waterside | 126 | 41(32.5) | 85(67.5) | 0(0) | 2(1.6) | 6(4.8) | 55(43.7) | 54(42.9) | 9(7.1) |
| 17 | Remo north | 22 | 7(31.8) | 15(68.2) | 0(0) | 0(0) | 1(4.5) | 10(45.5) | 9(40.9) | 2(9.1) |
| 18 | Shagamu | 168 | 30(17.9) | 138(82.1) | 1(0.6) | 21(12.5) | 12(7.1) | 46(27.4) | 80(47.6) | 8(4.8) |
| 19 | Egbado north | 156 | 99(63.5) | 57(36.5) | 4(2.6) | 5(3.2) | 13(8.3) | 42(26.9) | 83(53.2) | 9(5.8) |
| 20 | Egbado south | 95 | 35(36.8) | 60(63.2) | 0(0) | 5(5.3) | 6(6.3) | 28(29.5) | 54(56.8) | 2(2.1) |
| | **Total** | **1499** | **600(40.0)** | **899(60.0)** | **14(0.9)** | **161(10.7)** | **156(10.4)** | **454(30.3)** | **609(40.6)** | **105(7.0)** |

*Ascaris and Trichuris* co-distribution was recorded in 7 LGAs; Egbado north, Egbado south, Ado-odo ota, Ijebu east, Ipokia, Obafemi owode and Ogun waterside, while combination of *Trichuris* and hookworm was recorded in 6 LGAs; Ado-odo ota, Ipokia, Obafemi owode, Ogun waterside, Egbado north and Egbado south. Triple infections comprising *Ascaris*, *Trichuris* and hookworm was recorded in 6 LGAs (Table 2).

## Aggregated prevalence of *Ascaris*, *Trichuris* and hookworm infections in Ogun State

An aggregated prevalence of 17.2% was recorded for at least one infection of *Ascaris lumbricoides*, *Trichuris trichiura* or hookworm. The prevalence range was between 5.3–49.2% across the LGAs. The highest prevalence was recorded in Egbado South while the lowest was recorded in Egbado North, however, no infection was recorded in Ijebu-Ode (Fig 3). Of the 19 endemic LGAs, 9 had prevalence status ranging between 20.0%-49.2% and 10 LGAs had prevalence status ranging between 5.3%-15.8%. There were significant differences in the prevalence record for any STH species across the 19 endemic LGAs (P = 0.000) (Fig 4).

## Specific prevalence estimates for *Ascaris*, *Trichuris* and hookworm infections in Ogun State

By species' prevalence, an overall prevalence of 13.6% was recorded for *Ascaris lumbricoides*, followed by hookworm with 4.6% and *Trichuris trichiura* with 1.7% (Table 2). The lowest prevalence for the three infections were observed in Egbado north., while Abeokuta south, Obafemi

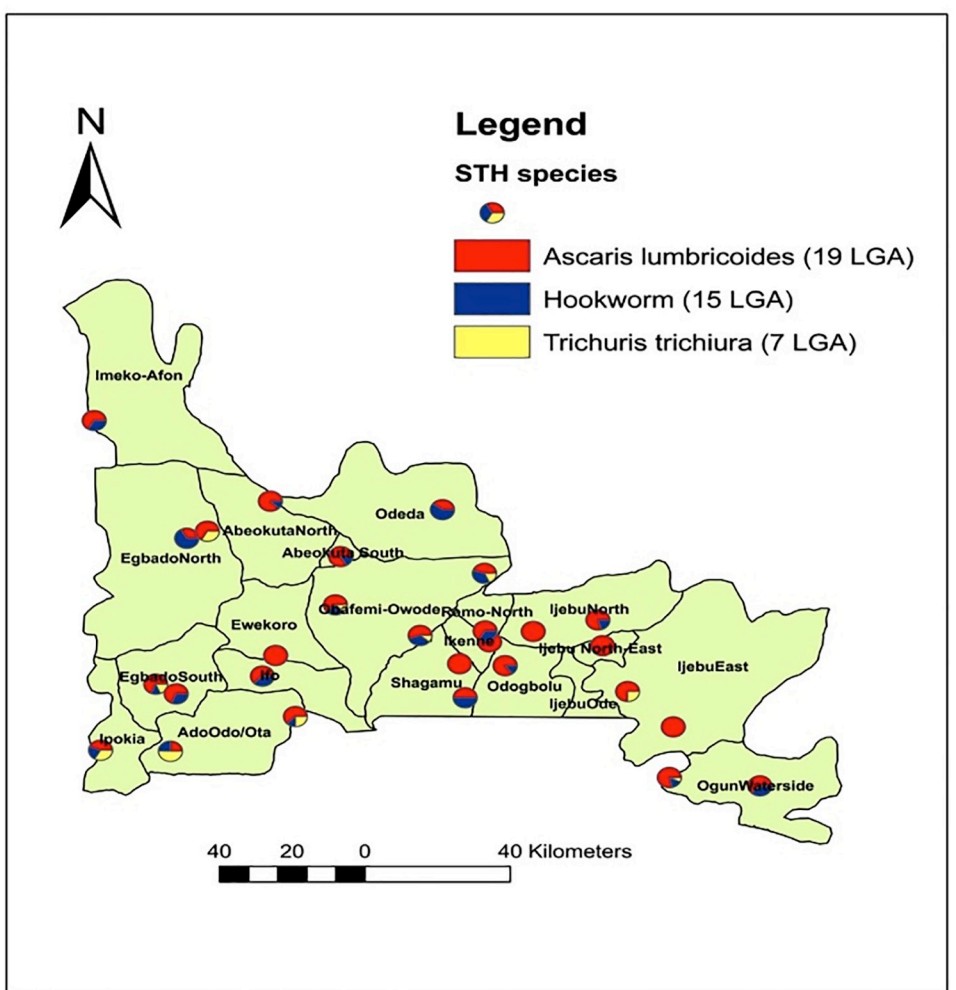

**Fig 1. Spatial distribution of *Ascaris, Trichuris* and hookworm infections in Ogun State.** Source: The authors using their primary data in ArcGIS software created this map. Permission: The authors give permission to re-use this map.

owode and Ipokia recorded the highest prevalence for ascariasis, hookworm infections and trichuriasis respectively (Table 2). Spatial-wise, *Ascaris lumbricoides was* the most predominant, found in 28(84.8%) communities (Fig 5), followed by hookworm in 19 (57.6%) communities (Fig 6) and *Trichuris trichiura* in 9(27.3%) communities (Fig 7). There were significant differences in the prevalence record for each parasitic infection across the endemic LGAs (P = 0.000).

## Aggregated and specific mean intensity estimates for *Ascaris, Trichuris* and hookworm infections in Ogun State

Table 3 shows the intensity estimates for *Ascaris, Trichuris* and hookworm infections in Ogun state. The aggregated geometric mean intensity of infections was 0.14±0.01epg with mean intensity ranging from 0.03±0.01epg to 0.43±0.06epg across the LGAs. The aggregated intensity shows that worm loads were highest in Obafemi owode and lowest in Imeko-afon. By

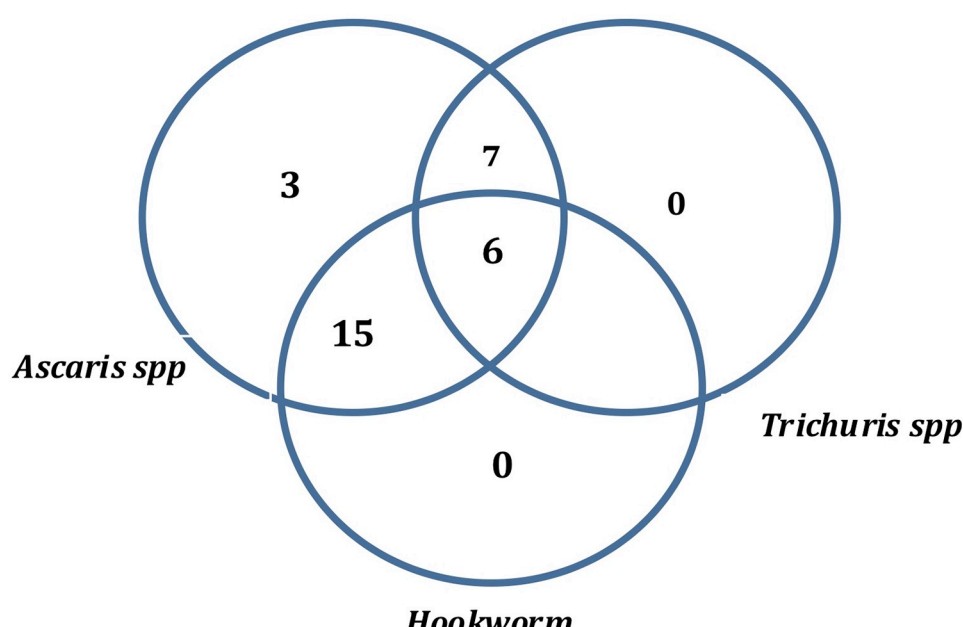

**Fig 2. Venn diagram showing the co-distribution of *Ascaris*, *Trichuris* and hookworm infections in Ogun State.** The numbers in the venn diagram reflects the number of affected LGAs. Source: The authors using their primary data in ArcGIS software created this map. Permission: The authors give permission to re-use this map.

**Table 2. Prevalence estimates for *Ascaris*, *Trichuris* and hookworm infections in Ogun State.**

| SN | LGA | NCE | NSE | A. lumbricoides NP (%) | T. Trichiura NP (%) | Hookworm NP (%) | Any STH NP (%) |
|---|---|---|---|---|---|---|---|
| 1 | Abeokuta North | 1 | 40 | 7(17.5) | 0(0) | 1(2.5) | 8(20.0) |
| 2 | Abeokuta South | 1 | 14 | 5(37.5) | 0(0) | 1(7.1) | 6(42.9) |
| 3 | Ado-odo ota | 2 | 56 | 6(10.7) | 4(7.1) | 2(3.6) | 8(14.3) |
| 4 | Ewekoro | 1 | 46 | 3(6.5) | 0(0) | 0(0) | 3(6.5) |
| 5 | Ifo | 1 | 19 | 5(26.3) | 0(0) | 3(15.8) | 7(36.8) |
| 6 | Ijebu East | 2 | 38 | 6(15.8) | 1(2.6) | 0(0) | 6(15.8) |
| 7 | Ijebu North | 2 | 48 | 6(12.5) | 0(0) | 1(2.1) | 6(12.5) |
| 8 | Ijebu North-East | 1 | 26 | 4(15.4) | 0(0) | 0(0) | 4(15.4) |
| 9 | Ijebu Ode | 1 | 25 | 0(0) | 0(0) | 0(0) | 0(0) |
| 10 | Ikenne | 1 | 36 | 11(30.6) | 0(0) | 0(0) | 11(30.6) |
| 11 | Imeko-Afon | 2 | 86 | 4(4.7) | 0(0) | 2(2.3) | 5(5.8) |
| 12 | Ipokia | 1 | 13 | 4(30.8) | 3(23.1) | 2(15.4) | 5(38.5) |
| 13 | Obafemi-Owode | 3 | 54 | 15(27.8) | 4(7.4) | 9(16.7) | 22(40.7) |
| 14 | Odeda | 2 | 71 | 4(5.6) | 0(0) | 6(8.5) | 8(11.3) |
| 15 | Odogbolu | 1 | 54 | 6(11.1) | 0(0) | 1(1.9) | 7(13.0) |
| 16 | Ogun waterside | 2 | 85 | 16(18.8) | 1(1.2) | 5(5.9) | 20(23.5) |
| 17 | Remo North | 1 | 15 | 4(26.7) | 0(0) | 2(13.3) | 6(40.0) |
| 18 | Shagamu | 2 | 128 | 9(7.0) | 0(0) | 3(2.3) | 10(7.8) |
| 19 | Egbado North | 4 | 114 | 3(2.6) | 1(0.9) | 2(1.8) | 6(5.3) |
| 20 | Egabdo South | 2 | 59 | 22(37.3) | 3(5.1) | 7(11.9) | 29(49.2) |
| | Total | 33 | 1027 | 140(13.6) | 17(1.7) | 47(4.6) | 177(17.2) |
| | Df | | | 19 | 19 | 19 | 19 |
| | Chi-square | | | 95.325 | 73.24 | 54.68 | 131.097 |
| | P-value | | | 0.00 | 0.00 | 0.00 | 0.00 |

NCS = number of communities sampled, NSE = number of stools examined, NP = number positive Df = Degree of freedom.

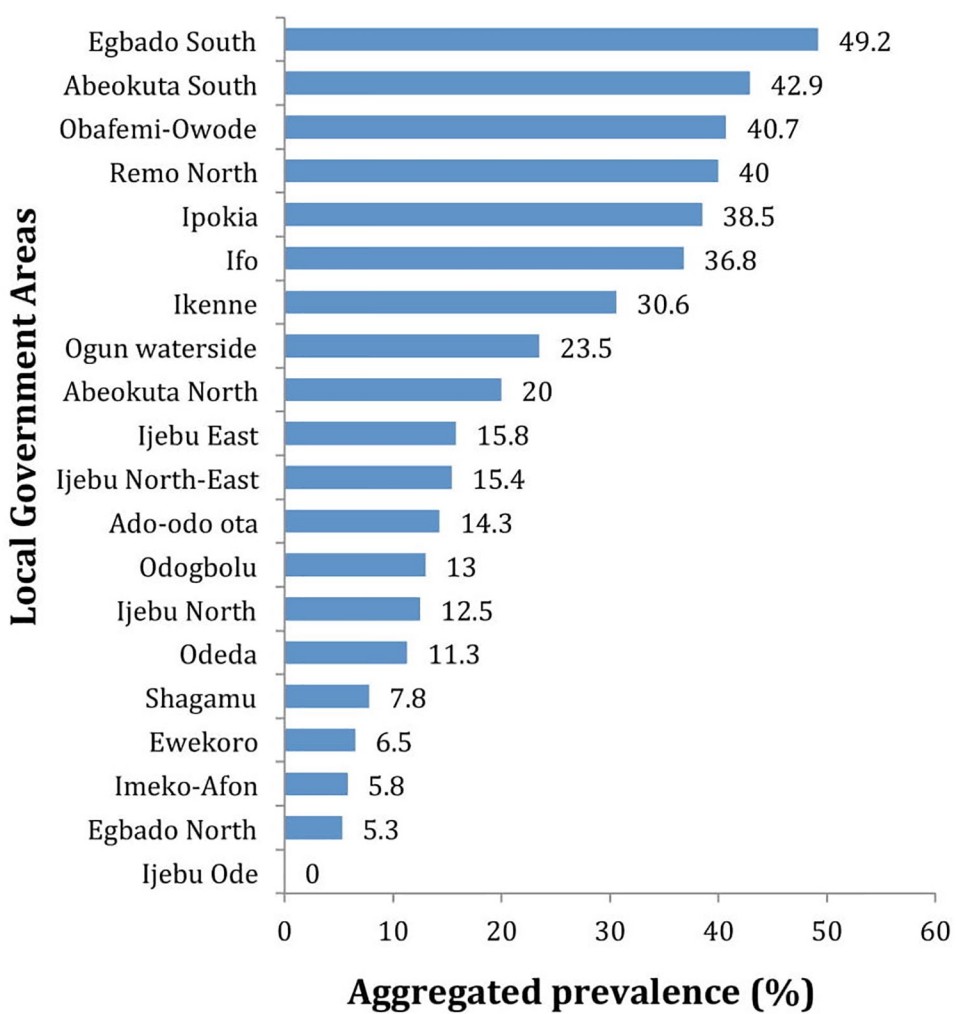

**Fig 3. Aggregated prevalence of *Ascaris*, *Trichuris* and hookworm infections in Ogun State.** Source: The authors using their primary data in ArcGIS software created this map. Permission: The authors give permission to re-use this map.

species intensities, *Ascaris lumbricoides* infection intensity was the highest with, 0.11±0.01epg, followed by hookworm with 0.03±0.01epg and *Trichuris trichiura* with 0.01±0.00epg. *Ascaris* mean intensities range between 0.01±0.01epg and 0.32±0.05epg, with the lowest in Imeko afon and highest in Ifo and Egbado south. For Hookworm infection, intensities range between 0.01±0.00epg and 0.13±0.04epg, with the lowest in Ijebu north, Egbado north, Egbado south, Ogun waterside, Imeko afon and highest in Obafemi-owode. *Trichuris* mean intensities range between 0.01±0.01epg and 0.10±0.05epg with the highest load in Ipokia and the lowest in Ogun waterside, Ijebu east and Egbado north.

## Prevalence of *Ascaris*, *Trichuris* and hookworm infections infection by sex and age characteristics

Table 4 shows the prevalence of infection by sex and age distribution of study participants. Majority of those infected were females 122(18.8%), although there were no significant

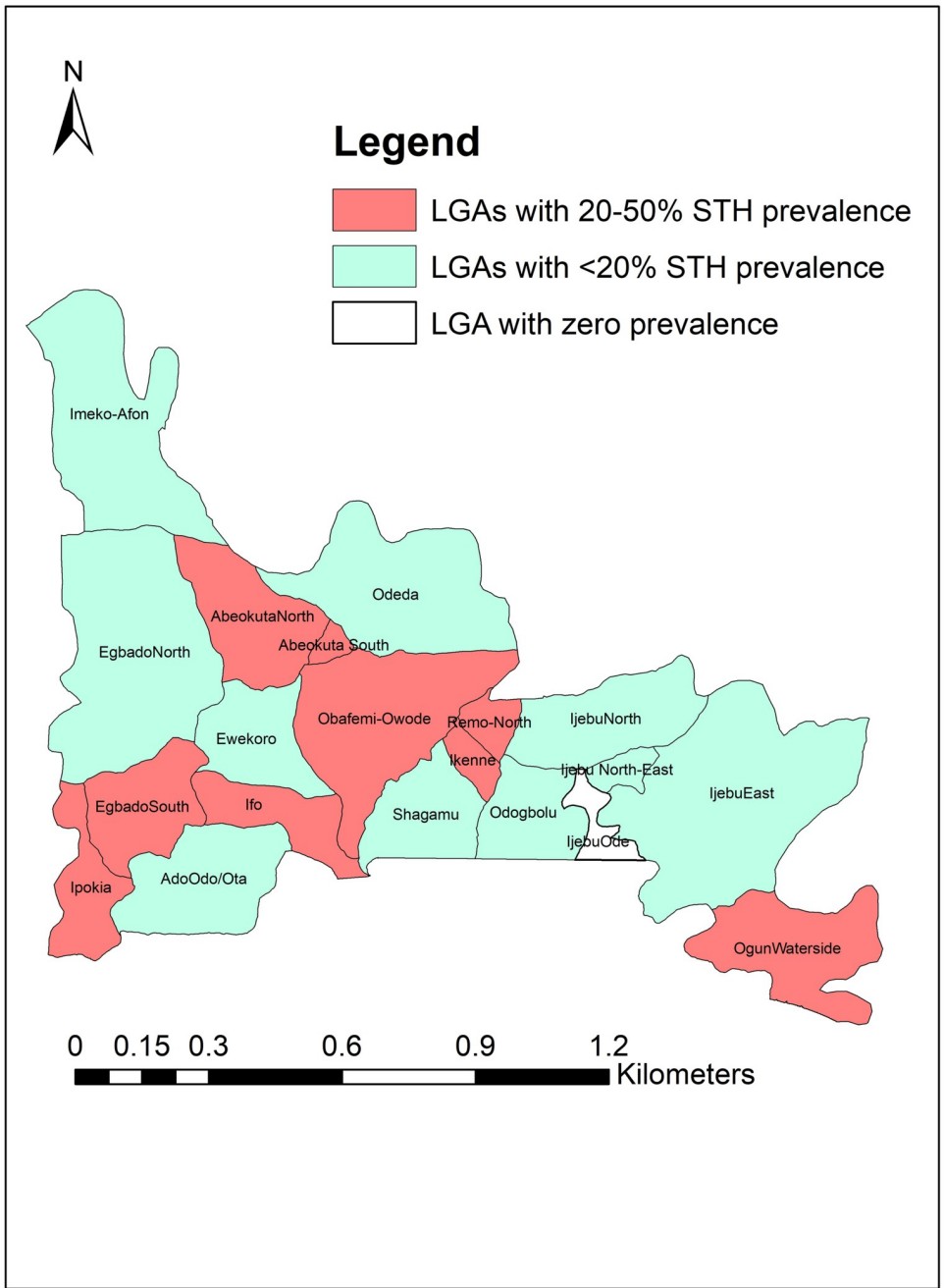

**Fig 4. LGAs classification by WHO prevalence thresholds for preventive chemotherapy.** Source: The authors using their primary data in ArcGIS software created this map. Permission: The authors give permission to re-use this map.

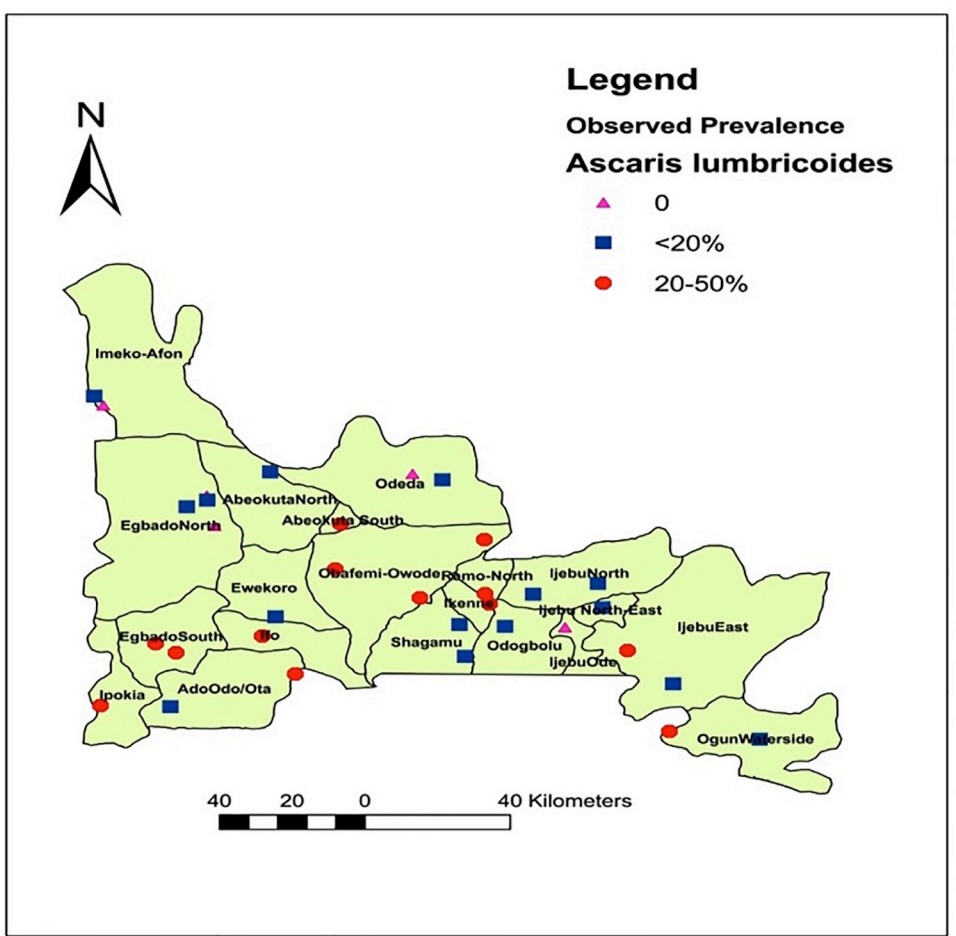

**Fig 5. Spatial distribution of *Ascaris lumbricoides* infection in Ogun State, Nigeria.** Source: The authors using their primary data in ArcGIS software created this map. Permission: The authors give permission to re-use this map.

differences in prevalence estimates across sex (P > 0.05). By age category, majority of those infected were adults and with age above 26 years, however there were no significant differences in prevalence of infection across age category (P > 0.05).

## Discussion

In Nigeria, research findings on the geospatial distribution of *Ascaris*, *Trichuris* and Hookworm infections are scarce, but emerging [11,12,13]. This study therefore adds to the wealth of existing epidemiological data on the prevalence and distribution of these infections in Ogun State, Nigeria. However, no attempt was made to analyse for risk factors. The overall prevalence recorded in this study (17.2%) is in line with the predictions of 13.8% and <20% prevalence reported by Oluwole *et al.* [11] and Yaro *et al.* [13] respectively. The prevalence values of the three soil transmitted helminth (STH) infections recorded in this study is comparable with the findings of Mogaji *et al.* [2], Oluwole *et al.* [12] and Gemechu et al. [14]. The similarities observed supports existing presumptions on decreasing trend of STH infections due to ongoing chemotherapy programmes in most endemic SSA countries [15,16]. However,

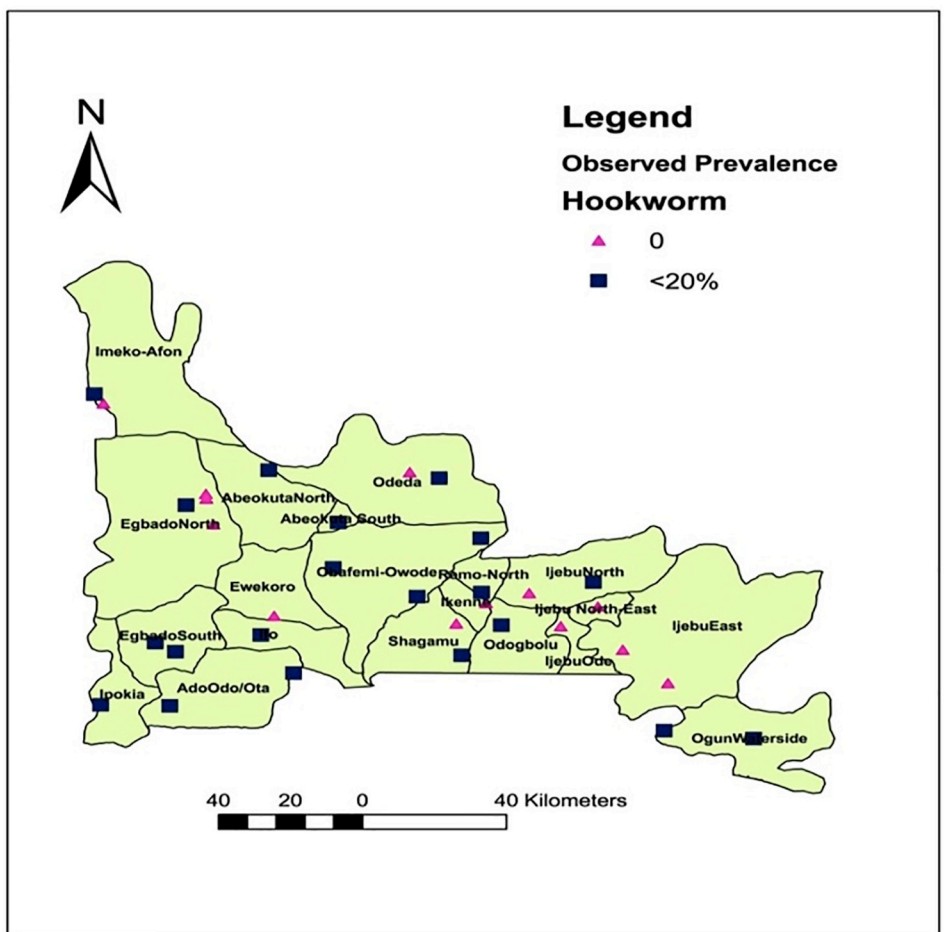

**Fig 6. Spatial distribution of hookworm infections in Ogun State, Nigeria.** Source: The authors using their primary data in ArcGIS software created this map. Permission: The authors give permission to re-use this map.

aggregated estimates at the state-level may unnecessarily mask the spatial patterns of disease distribution at LGA and communities, hence misleading intervention programmes and public health decisions. This peculiar dynamics of STH morbidity supports the need for intensified public health efforts in delineating localized hotspots and intervening appropriately to curtail the menace.

The prevalence recorded for *Ascaris lumbricoides* infections in this study is comparable with findings from previous epidemiological surveys [12,17]. The increase in prevalence of *A. lumbricoides infection* as observed in 19 of the 20 LGAs studied is similar to the findings of Oluwole et al [12] with respect to spatial pattern of *A. lumbricoides*. This may be due to similar ecological factors such as soil moisture, pH or temperature, which favours the transmission of STHs in the areas of study. The moderate prevalence of less than 20% recorded in various communities in contrast with prevalence of more than 50% reported by Oluwole et al. [12] may be due to the difference in the composition of population studied. Over 40% of the respondents in this study were adults within the age range 41-70years, as compared to 100% school-aged children population in the report of Oluwole *et al.* [12]. The school-aged children are known to be the most vulnerable to STH infections, and infections tend to be higher

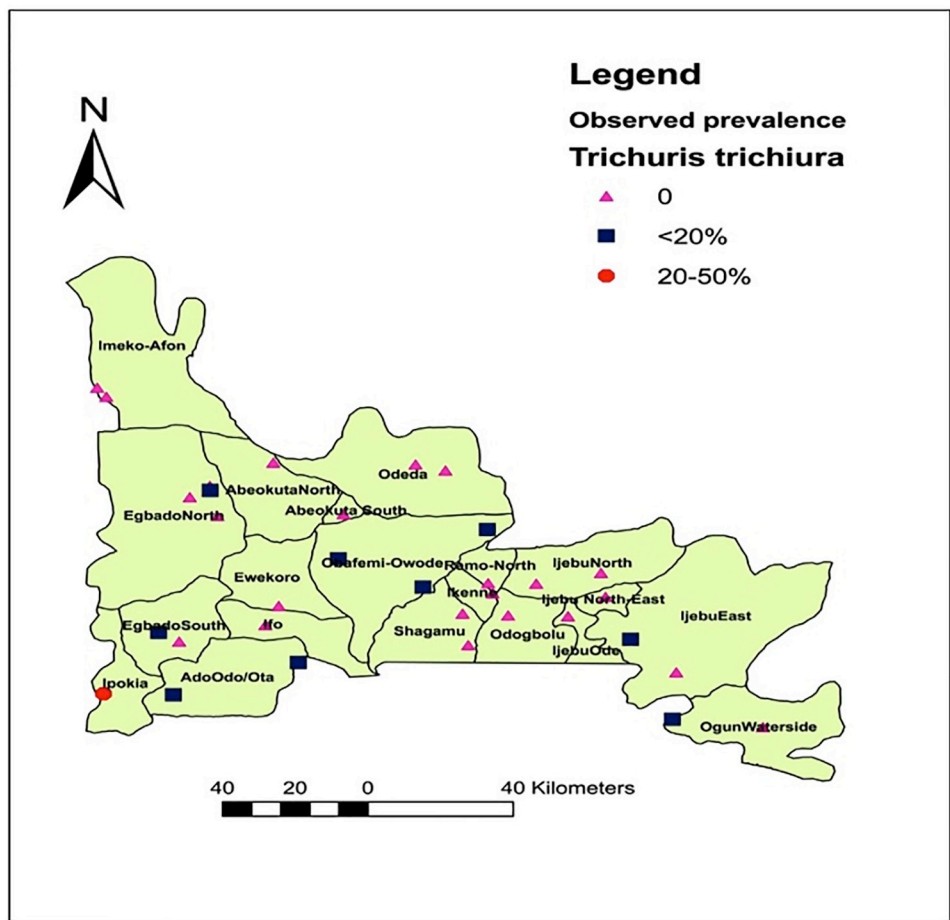

**Fig 7. Spatial distribution of *Trichuris trichiura* infections in Ogun State, Nigeria.** Source: The authors using their primary data in ArcGIS software created this map. Permission: The authors give permission to re-use this map.

among this group than any other subset in the community due to their risky behaviours that predisposes them to infection. Nevertheless, the findings from this study give a more complete population-based picture of current infection patterns that may exist in endemic communities, as infections may also cluster in households (adults) rather than school unit (children) only [18,19,20,21].

*Trichuris trichiura* has a restricted distribution in Ogun State. In fact, the prevalence of this parasite has been consistently reported to be low and non-existent in many parts of the country [2,11,12,13,17,22]. As such the low prevalence of 1.7% reported in this study is expected, and the geo-spatial restrictions of *Trichuris trichiura* ova to the southern part of the state may have been due to favourable environmental conditions [11]. In addition, there are similarities in the spatial distribution of *Trichuris* observed in this study with that reported by Oluwole *et al.* [12].

Hookworm infections is the second most common infection recorded in this study, with peak prevalence's reported among the adult respondents. This observation is in line with already established fact that prevalence and intensities of hookworm infections are restricted to adult populations [22]. However, the prevalence recorded in this study (4.6%) is lower than

**Table 3. Intensity (Mean ± SEM) of *Ascaris, Trichuris* and hookworm infections in selected population in Ogun State.**

| SN | LGA | NCS | NSE | *A. lumbricoides* Mean epg ± SEM | *T. Trichiura* Mean epg ± SEM | Hookworm Mean epg ± SEM | Any STH Mean epg ± SEM |
|---|---|---|---|---|---|---|---|
| 1 | Abeokuta north | 1 | 40 | 0.19±0.08 | 0.00±0.00 | 0.03±0.03 | 0.22±0.08 |
| 2 | Abeokuta south | 1 | 14 | 0.18±0.07 | 0.00±0.00 | 0.00±0.00 | 0.21±0.07 |
| 3 | Ado-odo ota | 2 | 56 | 0.09±0.03 | 0.00±0.00 | 0.05±0.02 | 0.14±0.05 |
| 4 | Ewekoro | 1 | 46 | 0.05±0.03 | 0.00±0.00 | 0.00±0.00 | 0.05±0.03 |
| 5 | Ifo | 1 | 19 | 0.32±0.13 | 0.00±0.00 | 0.12±0.07 | 0.41±0.13 |
| 6 | Ijebu east | 2 | 38 | 0.10±0.04 | 0.01±0.01 | 0.00±0.00 | 0.11±0.04 |
| 7 | Ijebu north | 2 | 48 | 0.09±0.04 | 0.00±0.00 | 0.01±0.01 | 0.09±0.04 |
| 8 | Ijebu north east | 1 | 26 | 0.20±0.11 | 0.00±0.00 | 0.00±0.00 | 0.20±0.11 |
| 9 | Ijebu ode | 1 | 25 | 0.00±0.00 | 0.00±0.00 | 0.00±0.00 | 0.00±0.00 |
| 10 | Ikenne | 1 | 36 | 0.20±0.60 | 0.00±0.00 | 0.00±0.00 | 0.20±0.06 |
| 11 | Imeko-afon | 2 | 86 | 0.01±0.01 | 0.00±0.00 | 0.01±0.01 | 0.03±0.01 |
| 12 | Ipokia | 1 | 13 | 0.15±0.07 | 0.10±0.05 | 0.04±0.03 | 0.23±0.09 |
| 13 | Obafemi-owode | 3 | 54 | 0.31±0.07 | 0.04±0.02 | 0.13±0.04 | 0.45±0.07 |
| 14 | Odeda | 2 | 71 | 0.07±0.03 | 0.00±0.00 | 0.10±0.04 | 0.14±0.05 |
| 15 | Odogbolu | 1 | 54 | 0.04±0.01 | 0.00±0.00 | 0.02±0.02 | 0.06±0.02 |
| 16 | Ogun waterside | 2 | 85 | 0.14±0.03 | 0.01±0.01 | 0.01±0.01 | 0.15±0.03 |
| 17 | Remo north | 1 | 15 | 0.14±0.06 | 0.00±0.00 | 0.04±0.02 | 0.18±0.06 |
| 18 | Shagamu | 2 | 128 | 0.05±0.02 | 0.00±0.00 | 0.01±0.01 | 0.06±0.02 |
| 19 | Egbado north | 4 | 114 | 0.02±0.01 | 0.01±0.01 | 0.01±0.00 | 0.04±0.01 |
| 20 | Egbado south | 2 | 59 | 0.32±0.05 | 0.04±0.02 | 0.08±0.03 | 0.43±0.06 |
| | **Total** | **33** | **1027** | **0.11±0.01** | **0.01±0.00** | **0.03±0.01** | **0.14±0.01** |

NCS = number of communities sampled, NSE = number of stools examined, epg = egg per gram of faeces, SEM = Standard Error of Mean

**Table 4. Prevalence of STH infections in relation to sex and age in selected population.**

| LGA | NE | *A. lumbricoides* NP (%) | *T. Trichiura* NP (%) | Hookworm NP (%) | Any STH NP (%) |
|---|---|---|---|---|---|
| **Sex** | | | | | |
| Male | 430 | 50(11.6) | 9(2.1) | 18(4.2) | 65(15.1) |
| Female | 597 | 90(15.1) | 8(1.3) | 29(4.9) | 112(18.8) |
| Total | 1027 | 140(13.6) | 17(1.7) | 47(4.6) | 177(17.2) |
| P value | | 0.112 | 0.351 | 0.611 | 0.127 |
| Df | | 1 | 1 | 1 | 1 |
| Chi square | | 2.523 | 0.871 | 0.258 | 2.327 |
| **Age group in years** | | | | | |
| <5 | 11 | 2(18.2) | 0(0) | 0(0) | 2(18.2) |
| 5–15 | 110 | 15(13.6) | 0(0) | 3(2.7) | 17(15.5) |
| 16–25 | 101 | 11(10.9) | 2(2.0) | 2(2.0) | 13(12.9) |
| 26–40 | 317 | 45(14.2) | 7(2.2) | 18(5.7) | 61(19.2) |
| 41–70 | 422 | 57(13.5) | 7(1.7) | 22(5.2) | 74(17.5) |
| >70 | 66 | 10(15.2) | 1(1.5) | 2(3.0) | 10(15.2) |
| Total | 1027 | 140(13.6) | 17(1.7) | 47(4.6) | 177(17.2) |
| P value | | 0.958 | 0.745 | 0.469 | 0.743 |
| Df | | 5 | 5 | 5 | 5 |
| Chi square | | 1.058 | 2.705 | 4.582 | 2.723 |

Df = degree of freedom

those reported by Mogaji *et al.* [2]; Oluwole *et al.* [12] and Fafunwa *et al.* [23] where prevalence estimate within the range 7.1–26.2% was observed. Nevertheless, the prevalence is higher than the 4.13% reported by Sam-wobo *et al.* [16]. The disparities observed across these studies maybe due to differences in study area/population, seasonality, soil parameters, environmental variables and socioeconomic status. In addition, walking barefooted which is a major risk factor in the percutaneous transmission of hookworm parasites is a common feature of school-aged children. The low prevalence recorded for hookworm infections may therefore be attributed to the fact that school-aged children constitutes only 10% of the study population.

## Conclusion

This study has shown the distribution of *Ascaris*, *Trichuris* and hookworm infections in Ogun State, Nigeria. These results and maps are useful and can serve as decision support tools for targeting, planning and delivery of intervention programmes aimed at controlling STH morbidity.

## Supporting information

**S1 Dataset.**
(XLSX)

## Acknowledgments

We are grateful to the community leaders and spokespersons that assisted with entry into the study locations. We also appreciate the study participants for their co-operation and participation. Our appreciation also goes to the Ogun State Ministry of Health for their assistance with field approvals and permits.

## Author Contributions

**Conceptualization:** Hammed Oladeji Mogaji, Uwem Friday Ekpo.

**Data curation:** Hammed Oladeji Mogaji.

**Investigation:** Hammed Oladeji Mogaji, Samuel Bankole, Adejuwon Adeniji, Mariam Tobi Fagbenro, Olaitan Olamide Omitola, Nnayere Simon Odoemene, Uwem Friday Ekpo.

**Methodology:** Hammed Oladeji Mogaji, Gabriel Adewunmi Dedeke, Babatunde Saheed Bada, Samuel Bankole, Adejuwon Adeniji, Mariam Tobi Fagbenro, Olaitan Olamide Omitola, Akinola Stephen Oluwole, Nnayere Simon Odoemene, Chiedu Felix Mafiana, Uwem Friday Ekpo.

**Project administration:** Hammed Oladeji Mogaji.

**Software:** Hammed Oladeji Mogaji, Olaitan Olamide Omitola.

**Supervision:** Gabriel Adewunmi Dedeke, Babatunde Saheed Bada, Akinola Stephen Oluwole, Eniola Micheal Abe, Chiedu Felix Mafiana, Uwem Friday Ekpo.

**Writing – original draft:** Hammed Oladeji Mogaji, Gabriel Adewunmi Dedeke, Babatunde Saheed Bada, Mariam Tobi Fagbenro, Eniola Micheal Abe, Uwem Friday Ekpo.

**Writing – review & editing:** Hammed Oladeji Mogaji, Gabriel Adewunmi Dedeke, Babatunde Saheed Bada, Akinola Stephen Oluwole, Eniola Micheal Abe, Chiedu Felix Mafiana, Uwem Friday Ekpo.

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
