## [Decision Letter · Decision Letter 0]

25 Mar 2020

PONE-D-20-06434

Spatial Distribution of Ascariasis, Trichuriasis and Hookworm Infections in Ogun State, Southwestern Nigeria.

PLOS ONE

Dear Dr Mogaji,

Thank you for submitting your manuscript to PLOS ONE. After careful consideration, we feel that it has merit but does not fully meet PLOS ONE’s publication criteria as it currently stands. Therefore, we invite you to submit a revised version of the manuscript that addresses the points raised during the review process.

ACADEMIC EDITOR: The study results provided some useful information, the spatial distributions of STH could be evaluated in a better/clearer way. I suggest the author modify the spatial analysis and map presentations and re-submit the revised version. Attached, 2 independent reviewers have some suggestions for your consideration.

We would appreciate receiving your revised manuscript by May 09 2020 11:59PM. To enhance the reproducibility of your results, we recommend that if applicable you deposit your laboratory protocols in protocols.io, where a protocol can be assigned its own identifier (DOI) such that it can be cited independently in the future. For instructions see: http://journals.plos.org/plosone/s/submission-guidelines#loc-laboratory-protocols

We look forward to receiving your revised manuscript.

Kind regards,

Chia Kwung Fan, LL.M, PhD

Academic Editor

PLOS ONE

Journal Requirements:

2. 

We suggest you thoroughly copyedit your manuscript for language usage, spelling, and grammar. If you do not know anyone who can help you do this, you may wish to consider employing a professional scientific editing service.  

4. Please include your tables as part of your main manuscript and remove the individual files. Please note that supplementary tables (should remain/ be uploaded) as separate "supporting information" files

6. We note that [Figure(s) 1,2,5,6,7,8] in your submission contain [map] images which may be copyrighted. All PLOS content is published under the Creative Commons Attribution License (CC BY 4.0), which means that the manuscript, images, and Supporting Information files will be freely available online, and any third party is permitted to access, download, copy, distribute, and use these materials in any way, even commercially, with proper attribution. For these reasons, we cannot publish previously copyrighted maps or satellite images created using proprietary data, such as Google software (Google Maps, Street View, and Earth). For more information, see our copyright guidelines: http://journals.plos.org/plosone/s/licenses-and-copyright.

1.     You may seek permission from the original copyright holder of Figure(s) [1,2,5,6,7,8,] to publish the content specifically under the CC BY 4.0 license.  

Additional Editor Comments (if provided):

The study results provided some useful information, the spatial distributions of STH could be evaluated in a better/clearer way. I suggest the author modify the spatial analysis and map presentations and re-submit the revised version.

Reviewers' comments:

Reviewer's Responses to Questions

**Comments to the Author**

1. Is the manuscript technically sound, and do the data support the conclusions?

Reviewer #1: Yes

Reviewer #2: Yes

2. Has the statistical analysis been performed appropriately and rigorously? 

Reviewer #1: Yes

Reviewer #2: Yes

3. Have the authors made all data underlying the findings in their manuscript fully available?

Reviewer #1: No

Reviewer #2: No

4. Is the manuscript presented in an intelligible fashion and written in standard English?

Reviewer #1: Yes

Reviewer #2: Yes

5. Review Comments to the Author

Reviewer #1: The manuscript entitled “Spatial Distribution of Ascariasis, Trichuriasis and Hookworm Infections in Ogun State, Southwestern Nigeria” investigated STH infections in Ogun State and try to mapped the spatial distribution of STH infection. Though the study results provided some useful information, the spatial distributions of STH could be evaluated in a better/clearer way. I suggest the author modify the spatial analysis and map presentations and re-submit the revised version. Below are some comments:

Major issues:

1. It seems that the georeferenced household’s information was not analyzed in the study because the figure only showed the locations of the selected communities. Since you have more than thousand georeferenced households participated in your study, why don’t you apply interpolation techniques (e.g. kriging or kernel density) to evaluate the prevalence and intensity?

2. The range of different age groups are different (e.g. 16-25, 41-70), any reason for the grouping criteria? The age distributions could be biased.

3. The investigation has been done at community level; however, the results were still presented at LGA level, and it’s hard to identify where is the most prevalent area. For instance, I have no idea where is “Yewa South” on the map (the LGA with the highest prevalence rate). Please modify the map presentation in a better way.

4. This is a cross-sectional survey which might be affected by many other factors. For example, does the infection prevalence influence by any intervention activities? Such limitation should be discussed in the manuscript.

5. Though the overall prevalence is not very high in the study area, the transmission still varied by locations. Can you explain why “Yewa South” has the highest prevalence?

Minor issue:

1. The scale bar in Figure 1 is not correct. It should not be “millimeters.” Please check whether you apply a correct projection coordination system.

2. Table 1 can be moved to the supplementary materials.

3. Figure 5. What does “Ogun State” mean in the legend? For the legend indicated the LGA with no case, please add the border.

Reviewer #2: Specific comments

Abstract

Result section

Line 2-4: The sentences “Ascariasis was the most spatially distributed found in 28 communities, with 140(13.6%) infected subjects, followed byHookworm infection in 19 communities, with 47(4.6%) infected subjects and Trichuriasis in 9 communities, with 17(1.7%) infected subjects.”

should be reconstructed to read “ Ascariasis was the most frequently observed parasite in 28 communities with a prevalence of 13.6%, followed by hookworm with a prevalence of 4.6% while Trichuris was the least encountered with a prevalence of 1.7%.

Line 6-7: The sentence is incomplete “The highest and lowest distribution of what (?) was recorded in Yewa South and Yewa North respectively.

Line 8: Delete the phrase “By WHO preventive chemotherapy thresholds”

Conclusion

This study provides information on the prevalence and spatial risk of Ascariasis, Trichuriasis

and Hookworm infection that will serve as decision-support tool to help facilitate targeting of

control interventions.

The conclusion should be re-written as “ This study provides epidemiological data on the prevalence and spatial distribution of ascariasis , trichuriasis and hookworm infection which will add to the baseline data and guide the public health officers in providing appropriate control strategies in the endemic communities.

Background

Line 7-8: Replace the phrases in parentheses (children between age 2 and 5) and (children between age 5 and 14) with (age 2-5 years) and (age 5-14 years).

Line 15: The sentence “there is an established morbidity control programme targeted at school-aged children due to their high level of contact with soils” should be re-written as “there is an established morbidity control programme targeted at school-aged children due to high worm burden as a result of their frequent contact with contaminated soils, ……………………….

Methods

Study area

Line 10: The phrase “The major occupation of the population is……………… should be re-phrased as “Major occupations of the population are………………

Study design and sampling technique

Line 1: The sentence “This study was community-based and cross-sectional in design” should be re-written as “This is a cross-sectional and community-based study”.

Selection of households for survey

Line 2: The sentence “communities were visited and sensitized about the study procedures

and were openly invited to participate in the research through community meetings and town

announcers” should be replaced with “selected communities were visited and with the permission of the community leaders, meetings were held with the members of the community to intimate them about the purpose of the study and the procedures to be adopted”.

The sub title “Collection of faecal samples and parasitological diagnosis” be replaced with “ Faecal samples collections and processing”.

Line 3-4: The sentence “All participants were asked to provide a sufficiently large stool sample (at least 5 g)” should be replaced with “Individuals who consented to take part in the study were requested to provide stool up to half (about 5gm) into the labelled plastic bottles provided.

Line 6: Delete the word “Precisely”.

Line 7-8: The sentence “The bottle was covered and agitated vigorously to suspend the stool efficiently in the solution” be replaced with “The bottle was covered and shaken vigorously to form a cloudy suspension”.

Line 9: Delete the word “further”

Line 11: The sentence “The supernatant was also discarded after centrifuging” be replaced with “The supernatant was decanted”.

Line 12-14: The sentence “7 ml of normal saline was later added to the sediment, after which 3 ml of ether was finally added to the suspension”. ( I did not agree with this procedure, it is faulty). The deposit or residue obtained after decanting the supernatant should be resuspended in distilled water or saline of a known vol and shaking vigorously. Thereafter, an equal vol of ether will then be added (i.e if 3ml of distilled water or saline was used to resuspend the residue, 3ml of ether should then be added). The authors should correct this procedure.

Results

Sex and age distribution of study participants

The sentence “A total of 1,499 respondents were subsequently enrolled after consent across the 33 communities and 20 LGAs. However, only 1,027 (68.5%) of them consented to the provision

of adequate stool samples (~5g) for laboratory analysis for parasites’ ova or larva. By

demography, 899(60.0%) were females and 600(40.0%) were males” should be rewritten as “ A total of 1499 participants comprising of 899 (60%) females and 600 (40%) males were enrolled across the 33 communities in 20LGAs out of which 1024 (68.5%) returned stool samples for laboratory analysis.

The sentence “The sex ratio of females to males was 3:2. In addition, the highest number of respondents were within the age range 41-70yrs (40.6%), followed by 26-40yrs (30.3%), 5-15yrs (10.7%), 16-25yrs (10.4%) and >70yrs (7.0%) (Table 2)” should be re-written as “The ages of the enrolled participants are provided in table 2.

Co-distribution patterns of Ascaris, Trichuris and Hookworm infections in Ogun State

Line1: Replace the words “mono infection” with single infection

Line 2: Delete the word “only”

Line 3: Replace the word “co-distribution” with the word “combination”

Line 5: Begin with combination of Ascaris and Trichuris and delete the word “co-distribution”

Line 6 & 7: Rewrite “while Trichuris and Hookworm co-distribution was recorded in 6 LGAs;” as “while combination of Trichuris and hookworm was……………………….”

Line 8-10: Rewrite “However, for triple infections, the three species (Ascaris, Trichuris and Hookworm) were co-distributed in 6 LGAs; Ado-Odo, Ipokia, Obafemi-Owode, Ogun waterside, Yewa North and Yewa South” as “ Triple infection comprising Ascaris, Trichuris and hookworm was recorded in 6 LGAs (table 3)”.

Aggregated prevalence of Ascaris, Trichuris and Hookworm infections in Ogun State

The sentences “The aggregated prevalence of 17.2% was recorded for at least one infection of Ascaris lumbricoides, Trichuris trichiura or hookworm. The prevalence range was between 5.3 –

49.2% across the LGAs (Figure 4). Infections were highest in Yewa south, lowest in Yewa

north and no infection was recorded in Ijebu-Ode. Of the 19 endemic LGAs, 9 had prevalence

status ranging between 20.0%-49.2% (Abeokuta north, Abeokuta south, Ifo, Ikenne, Ipokia,

Obafemi owode, Ogun waterside, Remo north and Yewa south” should be reduced to “ The highest prevalence of 49.2% was recorded in Yewa South while the lowest of 5.3% was recorded in Yewa North, however, no infection was recorded in Ijebu-Ode.

Delete this sentence “However, 10 LGAs haprevalence status ranging between 5.3%-15.8% which is below the WHO recommended thresholds for preventive chemotherapy (Ado-odo ota, Ewekoro, Ijebu east, Ijebu north, Ijebu north east, Imeko afon, Odeda, Odogbolu, Sagamu and Yewa north).

Specific prevalence estimates for Ascaris, Trichuris and Hookworm infections in Ogun

State.

The contents under this section is too wordy. The authors should highlight significant findings in terms of the most frequently encountered parasite in each LGA, the range of other parasites and if there is significant differences in prevalence of infection across the communities.

Aggregated and specific mean intensity estimates for Ascaris, Trichuris and Hookworm

infections in Ogun State.

The authors should express the intensity as eggs per gram (epg) of faeces i.e epg should be put in front of the mean intensity values in all cases.

The authors stated under this section about the single, doubleand triple infections, however there was no dat presented to show the prevalences of this infection. The table should be provided.

Discussion.

The authors should use simple grammar to discuss their findings.

6. PLOS authors have the option to publish the peer review history of their article (what does this mean?). If published, this will include your full peer review and any attached files.

Reviewer #1: No

Reviewer #2: No

---

## [Author Response · Author response to Decision Letter 0]

2 Apr 2020

PONE-D-20-06434

Spatial Distribution of Ascariasis, Trichuriasis and Hookworm Infections in Ogun State, Southwestern Nigeria.

PLOS ONE

ACADEMIC EDITOR: The study results provided some useful information, the spatial distributions of STH could be evaluated in a better/clearer way. I suggest the author modify the spatial analysis and map presentations and re-submit the revised version. Attached, 2 independent reviewers have some suggestions for your consideration.

Response: Thank you very much for this observation. We will like to inform you that the objective of this present study is to provide epidemiological data on STH morbidity in the state, using household data, rather than school-based data. These data are useful for assessing the impact of school-based STH control on community prevalence. Households data were grouped into community data, and then aggregated at LGA level. The LGA is the implementation unit for school-based preventive chemotherapy programme in the state. Spatial analysis of the household data are still under analysis, and it not part of this report. As such, not to mislead the reviewers, we have removed the word spatial from the title to reflect the current analysis.

Reviewer 1:

Reviewer #1: The manuscript entitled “Spatial Distribution of Ascariasis, Trichuriasis and Hookworm Infections in Ogun State, Southwestern Nigeria” investigated STH infections in Ogun State and try to mapped the spatial distribution of STH infection. Though the study results provided some useful information, the spatial distributions of STH could be evaluated in a better/clearer way. I suggest the author modify the spatial analysis and map presentations and re-submit the revised version. Below are some comments:

Major issues:

C1. It seems that the georeferenced household’s information was not analyzed in the study because the figure only showed the locations of the selected communities. Since you have more than thousand georeferenced households participated in your study, why don’t you apply interpolation techniques (e.g. kriging or kernel density) to evaluate the prevalence and intensity?

R1: Thank you very much for this observation. We will like to inform you that the objective of this present study is to provide epidemiological data on STH morbidity in the state, using household data, rather than school-based data. These data are useful for assessing the impact of school-based STH control on community prevalence. Households data were grouped into community data, and then aggregated at LGA level. The LGA is the implementation unit for school-based preventive chemotherapy programme in the state. Spatial analysis of the household data are still under analysis, and it not part of this report. As such, not to mislead the reviewers, we have removed the word spatial from the title to reflect the current analysis.

C2. The range of different age groups are different (e.g. 16-25, 41-70), any reason for the grouping criteria? The age distributions could be biased.

R2: We intended to understand the dynamics of STH morbidity that may exist among the different age classes, i.e. preschoolers (<5yrs), School-aged children (5-15years), 16-25years (post-primary), youth (26-40 years), 40-70 (Adults), and aged (>70 years). Only the age classes 5-15 years are under preventive chemotherapy in the State.

C3. The investigation has been done at community level; however, the results were still presented at LGA level, and it’s hard to identify where is the most prevalent area. For instance, I have no idea where is “Yewa South” on the map (the LGA with the highest prevalence rate). Please modify the map presentation in a better way.

R3: Yes, the investigations were done at household level in the selected communities. However, results were presented both at the community level and LGA level. 

For instance, the maps created (Figure 5, 6 and 7) show the spatial distribution of the parasites at the community level (i.e. the points on the map refers to communities)

However, the map (Figure 4) shows the aggregated prevalence of STH infection by LGA. 

To avoid the presentation of cumbersome tables, results presented in tables were summarized at the LGA level. The LGA level is quite important as it remains the focal point for planning/delivery of MDA in the state. 

In addition, we realized that Yewa-north and Yewa south are not reflected in the maps. This is because these LGAs are also known as Egbado North and Egbado South respectively. We have thus adjusted our text to reflect Egbado North and Egbado South.

C4. This is a cross-sectional survey which might be affected by many other factors. For example, does the infection prevalence influence by any intervention activities? Such limitation should be discussed in the manuscript.

R4: Thank you very much. We have included this limitation in our discussion (Line 273-275)

This study therefore adds to the wealth of existing epidemiological data on the prevalence and distribution of these infections in Ogun State, Nigeria. However, no attempt was made to analyse for risk factors.

C5. Though the overall prevalence is not very high in the study area, the transmission still varied by locations. Can you explain why “Yewa South” has the highest prevalence?

R5: The southern region of the state has been characterized to have favorable environmental conditions that allow STH parasites to thrive. This coupled with other socio-economic factors may have contributed to the high prevalence recorded. 

Minor issue:

C6: The scale bar in Figure 1 is not correct. It should not be “millimeters.” Please check whether you apply a correct projection coordination system.

R6: We have to removed Figure 1 since it is less informative. And more importantly we have four figures showing the study area already. Thank you.

C7: Table 1 can be moved to the supplementary materials.

R7: Thank you very much. We have moved it to supplementary materials as suggested.

C8: Figure 5. What does “Ogun State” mean in the legend? For the legend indicated the LGA with no case, please add the border.

R8: Thank you very much. We have removed Ogun State from the legend. We have also included the border for the LGA with zero prevalence. 

Reviewer 2 Comments:

Reviewer #2: Specific comments

Abstract

Result section

C1: Line 2-4: The sentences “Ascariasis was the most spatially distributed found in 28 communities, with 140(13.6%) infected subjects, followed by Hookworm infection in 19 communities, with 47(4.6%) infected subjects and Trichuriasis in 9 communities, with 17(1.7%) infected subjects.” should be reconstructed to read “ Ascariasis was the most frequently observed parasite in 28 communities with a prevalence of 13.6%, followed by hookworm with a prevalence of 4.6% while Trichuris was the least encountered with a prevalence of 1.7%.

R1: We have modified our text as suggested. Please find the improved text in between lines 45-48. Thanks 

C2: Line 6-7: The sentence is incomplete “The highest and lowest distribution of what (?) was recorded in Yewa South and Yewa North respectively.

R2: We have improved our text.

Line 49: The highest and lowest distribution of overall helminth infections was recorded in Egbado South and Egbado North respectively

C3: Line 8: Delete the phrase “By WHO preventive chemotherapy thresholds”

R3: The phrase has been deleted as suggested. 

Line 50-51: Nine regions had infection status between 20.0%-49.2%, while 10 regions had infection status between 5.3%-15.8%. 

Conclusion

C4: This study provides information on the prevalence and spatial risk of Ascariasis, Trichuriasis and Hookworm infection that will serve as decision-support tool to help facilitate targeting of control interventions.

The conclusion should be re-written as “ This study provides epidemiological data on the prevalence and spatial distribution of ascariasis , trichuriasis and hookworm infection which will add to the baseline data and guide the public health officers in providing appropriate control strategies in the endemic communities.

R4: We have modified our text as suggested. Please find the improved text in between lines 53-56. Thanks

Line 53-56: This study provides epidemiological data on the prevalence and spatial distribution of ascariasis , trichuriasis and hookworm infections which will add to the baseline data and guide the public health officers in providing appropriate control strategies in the endemic communities.

Background

C5: Line 7-8: Replace the phrases in parentheses (children between age 2 and 5) and (children between age 5 and 14) with (age 2-5 years) and (age 5-14 years).

R5: We have modified our text as suggested. Please find the improved text in between lines 66-67. Thanks

Line 66-67: Besides, about 267 million pre-schoolers (age 2-5 years) and 568 million school-aged children (age 5-14 years) who live in areas where these

C6: Line 15: The sentence “there is an established morbidity control programme targeted at school-aged children due to their high level of contact with soils” should be re-written as “there is an established morbidity control programme targeted at school-aged children due to high worm burden as a result of their frequent contact with contaminated soils, ……………………….

R6: We have modified our text as suggested. Please find the improved text in between lines 72-74. Thanks

Lines 72-74: Nonetheless, in most endemic settings, there is an established morbidity control programme targeted at school-aged children due to high worm burden as a result of their frequent contact with contaminated soils

Methods

Study area

C7: Line 10: The phrase “The major occupation of the population is……………… should be re-phrased as “Major occupations of the population are………………

R7: We have modified our text as suggested. Please find the improved text in between lines 96. Thanks

Line 96: Major occupations of the population are farming, timber logging, and trading.

Study design and sampling technique

C8: Line 1: The sentence “This study was community-based and cross-sectional in design” should be re-written as “This is a cross-sectional and community-based study”.

R8: We have modified our text as suggested. Please find the improved text in between lines 101-102. Thanks

Line 101-102: This is a cross-sectional and community-based study, conducted between July 2016 and November 2018.

Selection of households for survey

C9: Line 2: The sentence “communities were visited and sensitized about the study procedures and were openly invited to participate in the research through community meetings and town announcers” should be replaced with “selected communities were visited and with the permission of the community leaders, meetings were held with the members of the community to intimate them about the purpose of the study and the procedures to be adopted”.

R9: We have modified our text as suggested. Please find the improved text in between lines 113-115. Thank you so much.

Line 113-115: Prior to data collection, selected communities were visited and with the permission of the community leaders, meetings were held with the members of the community to intimate them about the purpose of the study and the procedures to be adopted.

C10: The sub title “Collection of faecal samples and parasitological diagnosis” be replaced with “ Faecal samples collections and processing”.

R10: We have modified our text as suggested. Please find the improved text in line 122. Thank you

Line 122: Faecal samples collections and processing 

C11: Line 3-4: The sentence “All participants were asked to provide a sufficiently large stool sample (at least 5 g)” should be replaced with “Individuals who consented to take part in the study were requested to provide stool up to half (about 5gm) into the labelled plastic bottles provided.

R11: We have modified our text as suggested. Please find the improved text in between lines 125-127. Thank you so much.

Lines 125-127: Individuals who consented to take part in the study were requested to provide stool up to half (about 5gm) into the labelled plastic bottles provided.

C12: Line 6: Delete the word “Precisely”.

R12: Deleted as suggested. Please check Line 129

C13: Line 7-8: The sentence “The bottle was covered and agitated vigorously to suspend the stool efficiently in the solution” be replaced with “The bottle was covered and shaken vigorously to form a cloudy suspension”.

R13: We have modified our text as suggested. Please find the improved text in between lines 130-131. Thank you so much.

Line 130-131: The bottle was covered and shaken vigorously to form a cloudy suspension. The stool suspension……

C14: Line 9: Delete the word “further”

R14: Deleted as suggested.

Line 131: The stool suspension was strained into…..

C15: Line 11: The sentence “The supernatant was also discarded after centrifuging” be replaced with “The supernatant was decanted”.

R15: We have modified our text as suggested. Please find the improved text in line 133. Thank you

Line 133: The supernatant was decanted

C16: Line 12-14: The sentence “7 ml of normal saline was later added to the sediment, after which 3 ml of ether was finally added to the suspension”. ( I did not agree with this procedure, it is faulty). The deposit or residue obtained after decanting the supernatant should be resuspended in distilled water or saline of a known vol and shaking vigorously. Thereafter, an equal vol of ether will then be added (i.e if 3ml of distilled water or saline was used to resuspend the residue, 3ml of ether should then be added). The authors should correct this procedure.

R16: Thank you for this explanation. However, we followed the procedure describe by Endriss et al. 2005. We have included the reference immediately after the text. 

Results

Sex and age distribution of study participants

C17: The sentence “A total of 1,499 respondents were subsequently enrolled after consent across the 33 communities and 20 LGAs. However, only 1,027 (68.5%) of them consented to the provision of adequate stool samples (~5g) for laboratory analysis for parasites’ ova or larva. By demography, 899(60.0%) were females and 600(40.0%) were males” should be rewritten as “ A total of 1499 participants comprising of 899 (60%) females and 600 (40%) males were enrolled across the 33 communities in 20LGAs out of which 1024 (68.5%) returned stool samples for laboratory analysis.

R17: We have modified our text as suggested. Please find the improved text in between lines 161-163. Thank you so much.

Line 161-163: A total of 1499 participants comprising of 899 (60%) females and 600 (40%) males were enrolled across the 33 communities in 20LGAs out of which 1024 (68.5%) returned stool samples for laboratory analysis.

C18: The sentence “The sex ratio of females to males was 3:2. In addition, the highest number of respondents were within the age range 41-70yrs (40.6%), followed by 26-40yrs (30.3%), 5-15yrs (10.7%), 16-25yrs (10.4%) and >70yrs (7.0%) (Table 2)” should be re-written as “The ages of the enrolled participants are provided in table 2.

R18: We have modified our text as suggested. Please find the improved text in between lines 130-131. Thank you so much.

Line 163-164: The ages of the enrolled participants are provided in table 1.

C19-24: Co-distribution patterns of Ascaris, Trichuris and Hookworm infections in Ogun State

C19: Line1: Replace the words “mono infection” with single infection

C20: Line 2: Delete the word “only”

C21: Line 3: Replace the word “co-distribution” with the word “combination”

C22: Line 5: Begin with combination of Ascaris and Trichuris and delete the word “co-distribution”

C23: Line 6 & 7: Rewrite “while Trichuris and Hookworm co-distribution was recorded in 6 LGAs;” as “while combination of Trichuris and hookworm was……………………….”

C24: Line 8-10: Rewrite “However, for triple infections, the three species (Ascaris, Trichuris and Hookworm) were co-distributed in 6 LGAs; Ado-Odo, Ipokia, Obafemi-Owode, Ogun waterside, Yewa North and Yewa South” as “ Triple infection comprising Ascaris, Trichuris and hookworm was recorded in 6 LGAs (table 3)”.

R19-24: We have considered all the corrections (C19-24) and modified our text accordingly to reflect them. Please find the improved text in between lines 173-182. Thank you so much

Line 173-182:

Co-distribution patterns of Ascaris, Trichuris and Hookworm infections in Ogun State

Figure 3 shows the co-distribution patterns of STH infections in the State. Single infection was recorded for Ascaris lumbricoides in 3 LGAs; Ewekoro, Ikenne and Ijebu northeast. For double infections, combination of Ascaris and Hookworm were the most predominant, observed in 15 LGAs except Ewekoro, Ijebu east, Ijebu northeast, Ikenne and Ijebu ode. Combination of Ascaris and Trichuris co-distribution was recorded in 7 LGAs; Yewa north, Yewa south, Ado-odo ota, Ijebu east, Ipokia, Obafemi owode and Ogun waterside, while combination of Trichuris and hookworm was recorded in 6 LGAs; Ado-odo ota, Ipokia, Obafemi owode, Ogun waterside, Yewa north and Yewa south. Triple infections comprising Ascaris, Trichuris and hookworm was recorded in 6 LGAs (Table 3). 

Aggregated prevalence of Ascaris, Trichuris and Hookworm infections in Ogun State

C25: The sentences “The aggregated prevalence of 17.2% was recorded for at least one infection of Ascaris lumbricoides, Trichuris trichiura or hookworm. The prevalence range was between 5.3 – 49.2% across the LGAs (Figure 4). Infections were highest in Yewa south, lowest in Yewa north and no infection was recorded in Ijebu-Ode. Of the 19 endemic LGAs, 9 had prevalence

status ranging between 20.0%-49.2% (Abeokuta north, Abeokuta south, Ifo, Ikenne, Ipokia, Obafemi owode, Ogun waterside, Remo north and Yewa south” should be reduced to “ The highest prevalence of 49.2% was recorded in Yewa South while the lowest of 5.3% was recorded in Yewa North, however, no infection was recorded in Ijebu-Ode.

R25: We have considered the corrections and modified our text accordingly to reflect them. Please find the improved text in between lines 187-188. 

The highest prevalence was recorded in Yewa South while the lowest was recorded in Yewa North, however, no infection was recorded in Ijebu-Ode (Figure 4). 

C26: Delete this sentence “However, 10 LGAs haprevalence status ranging between 5.3%-15.8% which is below the WHO recommended thresholds for preventive chemotherapy (Ado-odo ota, Ewekoro, Ijebu east, Ijebu north, Ijebu north east, Imeko afon, Odeda, Odogbolu, Sagamu and Yewa north).

R26: We have considered the comments raised. We believe deleting this sentence will mask the pattern we are trying to showcase. Therefore we have modified our text accordingly to reflect them. Please find the improved text in between lines 188-191.

Of the 19 endemic LGAs, 9 had prevalence status ranging between 20.0%-49.2% and 10 LGAs had prevalence status ranging between 5.3%-15.8%. There were significant differences in the prevalence record for any STH species across the 19 endemic LGAs (P=0.000) (Figure 5).

Specific prevalence estimates for Ascaris, Trichuris and Hookworm infections in Ogun State.

C27: The contents under this section is too wordy. The authors should highlight significant findings in terms of the most frequently encountered parasite in each LGA, the range of other parasites and if there is significant differences in prevalence of infection across the communities.

R27: We have considered the suggestions raised and modified our text accordingly to reflect them. Please find the improved text in between lines 197-205

Specific prevalence estimates for Ascaris, Trichuris and Hookworm infections in Ogun State.

By species’ prevalence, an overall prevalence of 13.6% was recorded for Ascaris lumbricoides, followed by hookworm with 4.6% and Trichuris trichiura with 1.7% (Table 3). The lowest prevalence for the three infections were observed in Yewa north., while Abeokuta south, Obafemi owode and Ipokia recorded the highest prevalence for ascariasis, hookworm infections and trichuriasis respectively (Table 3). Spatial-wise, Ascaris lumbricoides was the most predominant, found in 28(84.8%) communities (Figure 6), followed by hookworm in 19 (57.6%) communities (Figure 7) and Trichuris trichiura in 9(27.3%) communities. There were significant differences in the prevalence record for each parasitic infection across the endemic LGAs (P=0.000). 

.

Aggregated and specific mean intensity estimates for Ascaris, Trichuris and Hookworm infections in Ogun State.

C28: The authors should express the intensity as eggs per gram (epg) of faeces i.e epg should be put in front of the mean intensity values in all cases.

R28: We have considered the suggestions raised and modified our text accordingly to reflect them. Please find the improved text in between lines 221-234

Line 221-234

Aggregated and specific mean intensity estimates for Ascaris, Trichuris and Hookworm infections in Ogun State.

Table 4 shows the intensity estimates for Ascaris, Trichuris and hookworm infections in Ogun state. The aggregated geometric mean intensity of infections was 0.14±0.01epg with mean intensity ranging from 0.03±0.01epg to 0.43±0.06epg across the LGAs. The aggregated intensity shows that worm loads were highest in Obafemi owode and lowest in Imeko-afon. By species intensities, Ascaris lumbricoides infection intensity was the highest with, 0.11±0.01epg, followed by hookworm with 0.03±0.01epg and Trichuris trichiura with 0.01±0.00epg. Ascaris mean intensities range between 0.01±0.01epg and 0.32±0.05epg, with the lowest in Imeko afon and highest in Ifo and Yewa south. For Hookworm infection, intensities range between 0.01±0.00epg and 0.13±0.04epg, with the lowest in Ijebu north, Yewa north, Yewa south, Ogun waterside, Imeko afon and highest in Obafemi-owode. Trichuris mean intensities range between 0.01±0.01epg and 0.10±0.05epg with the highest load in Ipokia and the lowest in Ogun waterside, Ijebu east and Yewa north. 

C31: The authors stated under this section about the single, double and triple infections, however there was no dat presented to show the prevalences of this infection. The table should be provided.

R31: A venn diagram was used to present the number of LGAs where single, double and triple infections could be found. Please see figure 3. We didn’t report specific combination prevalences in our text to reduce the number of tables.

Discussion.

C32: The authors should use simple grammar to discuss their findings.

R32: We have re-worked our discussion to enhance clarity. Please check Line 273-286

---

## [Decision Letter · Decision Letter 1]

27 Apr 2020

PONE-D-20-06434R1

Distribution of Ascariasis, Trichuriasis and Hookworm Infections in Ogun State, Southwestern Nigeria.

PLOS ONE

Dear Dr Mogaji,

Thank you for submitting your manuscript to PLOS ONE. After careful consideration, we feel that it has merit but does not fully meet PLOS ONE’s publication criteria as it currently stands. Therefore, we invite you to submit a revised version of the manuscript that addresses the points raised during the review process.

ACADEMIC EDITOR: Please revise the ms according the reviewer's suggestions to polish your ms.

We would appreciate receiving your revised manuscript by Jun 11 2020 11:59PM. To enhance the reproducibility of your results, we recommend that if applicable you deposit your laboratory protocols in protocols.io, where a protocol can be assigned its own identifier (DOI) such that it can be cited independently in the future. For instructions see: http://journals.plos.org/plosone/s/submission-guidelines#loc-laboratory-protocols

We look forward to receiving your revised manuscript.

Kind regards,

Chia Kwung Fan, LL.M, PhD

Academic Editor

PLOS ONE

Reviewers' comments:

Reviewer's Responses to Questions

**Comments to the Author**

1. If the authors have adequately addressed your comments raised in a previous round of review and you feel that this manuscript is now acceptable for publication, you may indicate that here to bypass the “Comments to the Author” section, enter your conflict of interest statement in the “Confidential to Editor” section, and submit your "Accept" recommendation.

Reviewer #1: All comments have been addressed

Reviewer #2: All comments have been addressed

2. Is the manuscript technically sound, and do the data support the conclusions?

Reviewer #1: Yes

Reviewer #2: Yes

3. Has the statistical analysis been performed appropriately and rigorously? 

Reviewer #1: Yes

Reviewer #2: Yes

4. Have the authors made all data underlying the findings in their manuscript fully available?

Reviewer #1: Yes

Reviewer #2: Yes

5. Is the manuscript presented in an intelligible fashion and written in standard English?

Reviewer #1: Yes

Reviewer #2: Yes

6. Review Comments to the Author

Reviewer #1: (No Response)

Reviewer #2: Comments on the revised version of the MS

General comments: The authors revised the MS in line with the suggestions provided, however, there are still some minor corrections that needs to be effected in the discussion section.

Specific comments

Discussion

Line 283-284: The prevalence estimates also corroborates ………………………………….due to chemotherapy programmes in SSA countries (The statement is not clear). However, I would suggest that the sentence should be re-written to read “The prevalence values for the 3STHs recorded in this study is comparable ………………………………. (The values obtained should be compared with the findings of previous studies and may then give reason for the similarities or differences.).

Line 287: Replace the words “Sub-state level” with LGAs and communities

Line 292: Replace the following words “reported” with “recorded”; “corroborates” with “comparable”; “recent” with “previous”.

Line 293-295: The sentence “Ascaris lumbricoides infections were observed in …………………………………..by Oluwole et al. (12) should be recast to read “The increase in prevalence of A. lumbricoides infection as observed in 19 of the 20 LGAs studied is similar to the findings of Oluwole et al (12) with respect to spatial pattern of A. lumbricoides. This may be due to similar ecological factors (mention the factors) which favours the transmission of STHs in the areas of study”.

Line 296-299: Please delete the sentences “where moderate……………………………………… of STH parasites in the state”.

Line 300-302: The sentences “However, none of the LGAs studied has……………………. Population in the report of Oluwole et al. (12)” should be recast to read “The moderate prevalence of less than 20% recorded in various communities in contrast with prevalence of more than 50% reported by Oluwole et al. (12) may be due to the difference in the composition of population studied.

Line 313: Please delete the word “aggregate”

Line 316: Your assertion that hookworms egg can withstand extreme temp (I guess you meant extreme heat). Remember hookworm eggs have thin shells compared with thick shell of Ascaris eggs. It is only Ascaris eggs which can withstand adverse environmental conditions.

Line 317: Please replace the word ‘commonalities” with the word “similarities”

Line 326: replace the word “observed” with “reported”

Line 332: Replace the word “study” with “studied”

Table 3: The title should be rephrased to read “Intensity (Mean ± SEM) of Ascaris, hookworm and Trichuris infection in selected population in Ogun State”.

In table 3: Re-write the abbreviation “SE” as “SEM”. Define SEM as Standard Error of Mean

Table 4: The title should be rephrased to read "Prevalence of STH infections in relation to sex and age in selected population

Still in Table 4: Under Age group insert the word "Years" and delete the word "Years" from each age group.

7. PLOS authors have the option to publish the peer review history of their article (what does this mean?). If published, this will include your full peer review and any attached files.

Reviewer #1: No

Reviewer #2: No

---

## [Author Response · Author response to Decision Letter 1]

28 Apr 2020

Reviewer #2: Comments on the revised version of the MS

General comments: The authors revised the MS in line with the suggestions provided, however, there are still some minor corrections that needs to be effected in the discussion section.

Specific comments

Discussion

C1: Line 283-284: The prevalence estimates also corroborates ………………………………….due to chemotherapy programmes in SSA countries (The statement is not clear). However, I would suggest that the sentence should be re-written to read “The prevalence values for the 3STHs recorded in this study is comparable ………………………………. (The values obtained should be compared with the findings of previous studies and may then give reason for the similarities or differences.).

R1: The corrections have been made as suggested. We have compared our prevalence values with three reports (2 from Nigeria and one from Ethiopia (Gemechu et al. 2020) 

Please find our improved text between Line 283-287

The prevalence values of the three soil transmitted helminth (STH) infections recorded in this study is comparable with the findings of Mogaji et al. [2], Oluwole et al. [12] and Gemechu et al. [14]. The similarities observed supports existing presumptions on decreasing trend of STH infections due to on-going chemotherapy programmes in most endemic SSA countries [15,16].

C2: Line 287: Replace the words “Sub-state level” with LGAs and communities

R2: This correction has been made (Line 289). Thank you 

C3: Line 292: Replace the following words “reported” with “recorded”; “corroborates” with “comparable”; “recent” with “previous”.

R3: This correction has been made.

Line 293-294: The prevalence recorded for Ascaris lumbricoides infections in this study is comparable with findings from previous epidemiological surveys [12,17].

C4: Line 293-295: The sentence “Ascaris lumbricoides infections were observed in …………………………………..by Oluwole et al. (12) should be recast to read “The increase in prevalence of A. lumbricoides infection as observed in 19 of the 20 LGAs studied is similar to the findings of Oluwole et al (12) with respect to spatial pattern of A. lumbricoides. This may be due to similar ecological factors (mention the factors) which favours the transmission of STHs in the areas of study”.

R4: The corrections have been made as suggested (Line 294-298)

The increase in prevalence of A. lumbricoides infection as observed in 19 of the 20 LGAs studied is similar to the findings of Oluwole et al [12] with respect to spatial pattern of A. lumbricoides. This may be due to similar ecological factors such as soil moisture, pH or temperature, which favours the transmission of STHs in the areas of study.

C5: Line 296-299: Please delete the sentences “where moderate……………………………………… of STH parasites in the state”.

R5: The sentences have been deleted. Thank you

C6: Line 300-302: The sentences “However, none of the LGAs studied has……………………. Population in the report of Oluwole et al. (12)” should be recast to read “The moderate prevalence of less than 20% recorded in various communities in contrast with prevalence of more than 50% reported by Oluwole et al. (12) may be due to the difference in the composition of population studied.

R6: The corrections have been made as suggested (Line 298-300)

The moderate prevalence of less than 20% recorded in various communities in contrast with prevalence of more than 50% reported by Oluwole et al. [12] may be due to the difference in the composition of population studied.

C7: Line 313: Please delete the word “aggregate”

R7: The word has been deleted. Thank you

C8: Line 316: Your assertion that hookworms egg can withstand extreme temp (I guess you meant extreme heat). Remember hookworm eggs have thin shells compared with thick shell of Ascaris eggs. It is only Ascaris eggs which can withstand adverse environmental conditions.

R8: Thank you for this comment. We have removed the line entirely to improve on clarity 

C9: Line 317: Please replace the word ‘commonalities” with the word “similarities”

R9: The corrections have been made as suggested (Line 313)

C10: Line 326: replace the word “observed” with “reported”

R10: The corrections have been made as suggested (Line 317)

C11: Line 332: Replace the word “study” with “studied”

R11: The corrections have been made as suggested

C12: Table 3: The title should be rephrased to read “Intensity (Mean ± SEM) of Ascaris, hookworm and Trichuris infection in selected population in Ogun State”.

R12: The corrections have been made as suggested (Line 253)

C13: In table 3: Re-write the abbreviation “SE” as “SEM”. Define SEM as Standard Error of Mean

R13: The corrections have been made as suggested (Line 253-256)

C14: Table 4: The title should be rephrased to read "Prevalence of STH infections in relation to sex and age in selected population

R14: The corrections have been made as suggested (Line 264-266)

C15: Still in Table 4: Under Age group insert the word "Years" and delete the word "Years" from each age group.

R15: The corrections have been made as suggested (Line 264-266)

---

## [Editor Report · Decision Letter 2]

6 May 2020

Distribution of Ascariasis, Trichuriasis and Hookworm Infections in Ogun State, Southwestern Nigeria.

PONE-D-20-06434R2

Dear Dr. Mogaji,

We are pleased to inform you that your manuscript has been judged scientifically suitable for publication and will be formally accepted for publication once it complies with all outstanding technical requirements.

With kind regards,

Chia Kwung Fan, LL.M, PhD

Academic Editor

PLOS ONE
---

## [Editor Report · Acceptance letter]

13 May 2020

PONE-D-20-06434R2 

Distribution of Ascariasis, Trichuriasis and Hookworm Infections in Ogun State, Southwestern Nigeria. 

Dear Dr. Mogaji:

I am pleased to inform you that your manuscript has been deemed suitable for publication in PLOS ONE. Congratulations! Your manuscript is now with our production department. 

With kind regards,

on behalf of

Dr. Chia Kwung Fan 

Academic Editor

PLOS ONE